# Brief communication: The role of geophysical imaging in local landslide early warning systems

Jim S. Whiteley[1,2], Arnaud Watlet[1], John Michael Kendall[3], Jonathan E. Chambers[1]

[1]Shallow Geohazards and Earth Observation, British Geological Survey, Environmental Science Centre, Nicker Hill, Nottingham, NG12 5GG, United Kingdom.
[2]School of Earth Sciences, University of Bristol, Wills Memorial Building, Queens Road, Bristol, BS8 1RJ, United Kingdom.
[3]University of Oxford, Department of Earth Sciences, South Parks Road, Oxford, OX1 3AN, United Kingdom

*Correspondence to*: J.S. Whiteley (jwhi@bgs.ac.uk)

**Abstract.** We summarise the contribution of geophysical imaging to local landslide early warning systems (LoLEWS), highlighting how LoLEWS design and monitoring components benefit from the enhanced spatial and temporal resolutions of time-lapse geophysical imaging. In addition, we discuss how with appropriate laboratory-based petrophysical transforms, these geophysical data can be crucial for future slope failure forecasting and modelling, linking other methods of remote sensing and intrusive monitoring across different scales. We conclude that in light of ever increasing spatiotemporal resolutions of data acquisition, geophysical monitoring should be a more widely considered technology in the toolbox of methods available to stakeholders operating LoLEWS.

## 1 Introduction

Landslide mitigation measures are broadly divided in to two types: engineering approaches to reduce frequency or intensity of failures; and vulnerability reduction measures that de-risk exposed elements (Pecoraro et al., 2019). Here we concentrate on the latter, through the use of landslide early warning systems (LEWS) for landslides triggered by increases in ground moisture (i.e., moisture in both soil and rock environments). LEWS are increasingly used to reduce vulnerability due to developments in supporting technology and databases, and because of their low cost of implementation and low impact on the environment. LEWS are commonly divided in to two groups: territorial landslide early warning systems (TeLEWS, also known as geographical landslide early warning systems) covering large areas at the catchment or multi-catchment scale and encompassing many vulnerable slopes (see Piciullo et al., 2018); and local landslide early warning systems (LoLEWS) (see Pecoraro et al., 2019) focusing on slope-scale early warning.

When acquiring information on a slope at risk of failure, desk studies, walkover surveys, remotely sensed data and local intrusive investigations of landslides tend to provide surface-only, or highly localised subsurface, information. Conceptual models of a landslide system inferred from these sources alone may lack spatial detail, leading to knowledge gaps that require interpolation across large volumes of the subsurface, or infilling with other data sources; it is in this latter capacity (i.e.,

characterisation of the subsurface) that geophysical imaging is most commonly applied to landslide investigation (Jongmans and Garambois, 2007). More recently, geophysical equipment has been adapted for long-term deployment to landslides, to remotely acquire time-lapse (i.e., monitoring) data that can be used to assess time-dependent properties affecting landslide stability. Many of these developments have taken place at natural landslide observatories, such as the Hollin Hill Landslide

Observatory in the UK. At such sites, environmental factors acting on short time scales (e.g., extreme precipitation events) and longer time scales (e.g., seasonal variations in ground moisture) can be monitored, imaged, and assessed using geophysical approaches at the whole-slope scale. The case studies arising from these types of long-term studies have been crucial in advancing the application of geophysical monitoring of unstable slopes (Whiteley et al., 2019).

Focusing on these concepts of geophysical characterisation and monitoring of landslides, we present a conceptual framework that highlights the role that geophysical imaging (supported by field and laboratory measurements) can play in establishing LoLEWS (Fig. 1) for landslides triggered by increases in ground moisture conditions. This framework uses a modified version of that proposed by Intrieri et al. (2013) for establishing a generic LoLEWS. Following this, we describe and summarise the major contributions that geophysical imaging can make to the LoLEWS components (and sub-components) presented in Fig.

1, which include: design (geological knowledge, risk scenarios, design criteria, choice of geo-indicators), monitoring (instruments installation, data collection, data transmission, data interpretation), forecasting (data elaboration, comparison with thresholds, forecasting methods, warning) and decision support (risk perception, safe behaviours, response to warning, stakeholder involvement) (after Intrieri et al., 2013). The conceptual framework (Fig. 1) also indicates some of the sources of uncertainty that may arise from the inclusion of geophysical data and geophysically-supported geotechnical measurements in

a LoLEWS. After summarising the contributions of geophysical imaging to these components, we consider the future of landslide characterisation and monitoring using geophysical imaging for LoLEWS, highlighting recent technological developments for acquiring and processing increasing spatiotemporal resolution geophysical data.

## 2 Geophysical imaging in LoLEWS

Geophysical methods in general, including electromagnetic, geoelectrical, gravitational, magnetic and seismic approaches,

provide data that are proxies for subsurface conditions related to lithological, hydrological or mechanical properties (Jongmans and Garambois, 2007). Subsurface imaging (i.e., tomographic) techniques comprising seismic (refraction, reflection and surface wave) and geoelectrical (resistivity) properties are particularly prevalent in landslide investigations due to their ability to provide structural information and proxy data of landslide conditions in two-, three- and four-dimensions (Whiteley et al., 2019), and it is these methods that we focus on in this work. Measurements of artificially-generated (i.e., active) signal sources

are made across surface-deployed sensor arrays, and then processed (i.e., inverted) to produce cross-sections or volumetric models of the subsurface. When repeat measurements are made at scheduled, regular intervals, time-lapse images are produced, revealing time-dependent, localised variations in soil and rock properties (see Whiteley et al., 2019 and references therein). A

complete set of measurements across an array of sensors can typically be acquired in hours or less (depending on the array size, terrain, and speed and density of measurements), and at daily or sub-daily (or longer) intervals. This is in contrast to near-continuous geophysical methods, which typically comprise arrays of autonomous single sensors (e.g., broadband seismometers) that record persistent and transient passive signals, making them well suited to long-term monitoring of kinematic processes associated with slope displacements. Although until recently near-continuous geophysical methods tended to possess much lower spatial resolutions than active-source monitoring arrays, broad-scale images can still be obtained from sparse sensor networks (see Whiteley et al., 2019 and references therein).

Therefore, geophysical imaging, particularly geoelectrical and seismic tomographic methods, can conceptually contribute to establishing LoLEWS, with particular benefits in the design and monitoring components due to the their ability to characterise and monitor slopes at high spatial resolutions. However, the imaging and visualisations provided by geophysics (in comparison to single point measurements from sensors, or from surface-only observations), and their ability to be calibrated with laboratory geotechnical measurements, brings benefits to other components of slope monitoring activities downstream, including forecasting and decision support. In our conceptual framework (Fig. 1), information from any downstream stage can be used to refine information gathered in upstream stages. For example, monitoring data may provide useful information to be incorporated in a detailed ground model of the landslide.

The framework also highlights the major sources of potential uncertainty that require consideration, including: i) laboratory testing of samples (e.g., quality of samples and how well samples reflect the heterogeneity of a geological formation) and issues scaling laboratory measurements to the slope-scale, ii) geophysical surveys, including the design of the survey (e.g., equipment used, data coverage and data resolution), the survey conditions (e.g., signal-to-noise ratio impacting measurement quality), uncertainties surrounding the inversion process (e.g., model sensitivity, fitting of geophysical models to measured data, or the sensitivity of the inversion to regularisation constraints), and the presence of unfavourable conditions for geophysical surveying (e.g., buried low-velocity or high conductivity zones inhibiting accurate seismic and geoelectrical measurements respectively), iii) the accuracy, precision and representativeness of non-geophysical point sensors used to supplement geophysical monitoring systems, and issues surrounding scaling localised point-sensor measurements to the slope-scale, iv) the identification of geophysical thresholds at which critical slope conditions are reached, v) the limitations, particularly in the ranges of co-sensitivity, of petrophysical relationships, and vi) the confidence of end-users to incorporate sources of geophysical data in to decision making processes surrounding slope-scale early warning. In the following sections we explore the emerging opportunities for integrating geophysical imaging into LoLEWS by summarising the contributions to the components in our framework (Fig. 1).

**2.1 Design**

**2.1.1 Geological knowledge**

Characterising the geological setting and identifying precursory conditions to landslide failure is an important initial step in establishing a LoLEWS (Intrieri et al., 2013). Geophysical imaging can help inform this stage by contributing to a spatially complete geoscientific understanding of the subsurface in terms of the geological setting, hydrological regime and geomorphological indicators of slope displacement. Characterisation and monitoring using geophysical imaging has a

100 demonstrably important role in establishing LoLEWS owing to high resolution spatiotemporal subsurface data acquisition, and the sensitivity of different methods to properties and processes that form and destabilise vulnerable slopes respectively (Whiteley et al., 2019). Geophysical measurements can be made at a range of depths and resolutions depending on financial and temporal constraints and study scope. The resolution of geophysical measurements made at the ground surface decrease with depth. Reconnaissance surveys, where measurements are acquired rapidly and at large measurement separations, offer

low-resolution data giving a broad overview of subsurface variations. Such reconnaissance information may guide the design of more detailed follow-up geophysical surveys, or may inform the design of additional intrusive investigations.

**2.1.2 Risk scenarios**

Geophysical data are well placed to offer slope-scale inputs for a range of qualitative (e.g., conceptual site model) and quantitative (e.g., analytical and numerical) modelling approaches. Quantitative analyses of slope failures using physical- or

110 process-based approaches are often required to model modes of failure, estimate potential mobilised volumes, predict landslide runout length and determine slope factor of safety, all of which will have a large effect on a given risk scenario. To mathematically model the stability of a slope, van Asch et al (2007) identified five key features on which information is required:

- Geometry (including topography), which can be obtained during geophysical survey deployment,

- Geomorphology, identified by geophysical variations indicating the presence of structural variations (e.g., slip surface(s), emplaced slipped material, surface fissures, etc.),

- Kinematics, such as landslide displacement rate and its controlling factors, which can be ascertained from geophysical monitoring,

- Geotechnics, where additional data from laboratory testing and the determination of petrophysical relationships (i.e.,

the estimation of a property such as porosity, density or moisture content from a proxy geophysical measurement) can provide appropriately discretised slope-scale geotechnical models (e.g., Uhlemann et al., 2017), and

- Geomechanics, which can be determined using outputs from the above step.

All of these types of models can be produced on discretised subsurface meshes, which can form inputs to the quantitative

monitoring of landslides.

### 2.1.3 Choice of geo-indicators

In a typical LoLEWS deployment, information on the environmental factors influencing or indicating displacement (or 'geo-indicators') are gathered from the installation of surface or subsurface sensors installed in the landslide. Geophysical models can be calibrated to a particular site condition (i.e., through petrophysical relationships, joint inversion or comparison with thresholds) after which the local condition measured at a sensor (or across a network of sensors) can be extrapolated and interpolated across a wide area at high resolution. It is recommended that measurement of one or several of a range of geo-indicators are made using associated instrumentation (Intrieri et al., 2013). This includes the direct use of geophysical measurements (in particular, geoelectrical measurements) as a geo-indicator of potential failure. The use of appropriate data elaboration techniques, typically linking geophysical and geotechnical measurements through laboratory based petrophysical relationship development, can convert other geophysical measurements to proxies for several other geo-indicators (e.g., seismic velocity to elastic moduli).

### 2.2 Monitoring

### 2.2.1 Instruments installation, data collection and data transmission

Geophysical monitoring applied to unstable slopes has advanced significantly in recent decades (Whiteley et al., 2019). Resistivity monitoring is one of the most developed geophysical methods for integrating in to LoLEWS, however, recent developments in seismic acquisition systems, such as the use of nodal arrays of seismic sensors or distributed acoustic sensing (DAS) systems are becoming increasingly prevalent. Several bespoke resistivity imaging systems have been developed for long-term deployment to unstable slopes, including ALERT (Kuras et al., 2009), GEOMON (Supper et al., 2012) and PRIME (Holmes et al., 2020) amongst others. In order to be suitable for slope monitoring, these systems have overcome several challenges, including the provision of off-grid power, the ability to automatically acquire scheduled measurements, and the inclusion of telemetry for near-real time transmission of data. For example, the PRIME resistivity monitoring system includes all of these elements, lending itself to integration with new and existing LoLEWS (Fig. 2). The low-cost modular and robust construction of monitoring systems such as PRIME, combined with their low-power consumption and optimised acquisition and data telemetry schedules make them adaptable for installing in harsh environments and difficult terrain, for example, where access may only be possible on foot (see Holmes et al., 2020; Whiteley et al., 2019 and references therein).

### 2.2.2 Data interpretation

Geophysical models require some specialist knowledge to interpret, and are often interpreted with a degree of uncertainty; but integrating multiple data streams (e.g., other geophysical methods, geotechnical observations, and remotely-sensed deformation data) increases the accuracy of the interpretation and reduces uncertainties (Whiteley et al. 2021). Inaccuracy and uncertainty can be ameliorated further with the use of petrophysical relationships described below. With this elaboration of geophysical data, each cell within the geophysical model can emulate the function of a geotechnical sensor, giving localised

information on subsurface properties, and in the case of geophysical monitoring data, time-series point information. As such, geophysical time-series data, from individual model cells (emulating point-source sensors), collections of model cells with similar properties (emulating geomorphological-scale features) or the entire model itself (considering the entire slope) can be incorporated with other sources of information collected at similar scales to identify thresholds at which failures may occur.

## 2.3 Forecasting

### 2.3.1 Data elaboration

A powerful approach to elaborating geophysical images is to link geophysical and geotechnical measurements in a quantitative manner through the use of petrophysical relationships (Fig. 3). Examples of this include Archie's equation (Archie, 1942) to determine porosity and saturation in clay-free rocks and soils, or the Waxman-Smits model for clay-rich material (Waxman and Smits, 1968) to translate resistivity measurements to gravimetric moisture content (e.g., Uhlemann et al., 2017). Other translations of resistivity include the use of relationships between the soil water potential, saturation and geoelectrical properties (Fredlund and Xing, 1994, Vanapalli et al., 1996) of a material to ultimately derive unsaturated shear strength from field resistivity measurements (e.g., Crawford and Bryson, 2018). Similar approaches can be applied to seismic data, where either estimations of density from field observations, or through laboratory measurements, can be combined with seismic velocity models to derive field-scale measurements of elastic moduli including bulk modulus, shear modulus, Young's modulus, and Poisson's ratio (e.g., Uhlemann et al., 2016). Alternatively, shear-strength and shear-wave velocity can be measured in the laboratory (e.g., using direct simple shear testing and bender element measurements) or in the field (e.g., using shear vanes and field seismic measurements) to derive a relationship between the two properties (e.g., Trafford and Long, 2020). Petrophysical joint inversion, where two geophysical datasets are inverted together to provide quantitative estimations of subsurface properties, are also possible when considering multiple co-located geophysical datasets, although this approach does not remove the need for non-geophysical observations to resolve some ambiguities arising in the resulting models (Wagner et al., 2019).

### 2.3.2 Comparison with thresholds / forecasting methods

Recently there has been a recognition that the use of subsurface data for establishing thresholds may improve LoLEWS, due to the effects of surface runoff, evapotranspiration, preferential flow and heterogeneous soil properties. For example, remotely sensed near-surface ground moisture data may reduce the number of false alarms in a LEWS compared to when only rainfall thresholds are used (Marino et al., 2020). Additionally, measurements of ground moisture content and/or deficit are increasingly being recognised by engineers as a potential indicator of slope failure. Geophysical imaging, particularly when translated to geotechnical or geomechanical models through petrophysical relationships, provides the opportunity to incorporate data at a range of scales and positions within the subsurface of a slope, for example, from installed geotechnical

sensors. Similarly, time-series geophysical monitoring data can provide inputs for machine-learning algorithms used in the nowcasting and forecasting of potential landslide failures.

## 2.4 Decision support

Using geophysical results for conveying complex spatial and temporal information is apt due to the scales and dimensions of the results in relation to the scale of unstable slopes. Geophysical data can easily be incorporated into 3D visualisation environments, where it can be displayed and manipulated alongside other sources of data in the development of integrated ground models (Whiteley et al. 2021). In LoLEWS, providing differential time-lapse images, or time-series of sub-sections of modelled data, can be an important step in translating the information from the technical to non-technical domain, and form

part of the visual basis for identifying slope instabilities or issuing warnings.

## 3 The future of geophysics for loLEWS

The continued integration of geophysical imaging approaches in to LoLEWS will be driven by developments in four areas: i) further research in to the petrophysical relationships between geophysical and geotechnical properties (in particular seismic velocity – stress state relationships) with improved consideration of uncertainty propagation through a LoLEWS, and with a

200 view to providing inputs to geophysical-geotechnical models of slope stability; ii) the maturation of technologies that allow the acquisition of passively acquired seismic data from fibre optic cables and large-n sensor arrays that can be deployed at the same spatial resolution as resistivity monitoring arrays; iii) the continued development of increasingly robust, low-power and low-cost geophysical systems for deployment to vulnerable slopes; iv) research in to the automation of data processing, modelling and interpretation, in order to streamline the ever increasing volumes of data being acquired from monitoring

systems. In addition, the establishment of a network of geophysically-supported LoLEWS within a catchment (or multi-catchment region) can also feed information in to TeLEWS, improving early warning at the larger scale. In this case, each LoLEWS site would be analogous to a network of sensors deployed within a single slope, each reporting the condition of their local area to establish a broader picture of landslide susceptibility based on near-real-time, slope-scale data.

Identifying appropriate slopes and research collaborators for the deployment of geophysics-supported LoLEWS is partly an issue of outreach, but one that is addressed through the establishment of interdisciplinary organisations such as the recently established LandAware: the international network on LEWS (Calvello et al., 2020). Additionally, the work undertaken at natural observatory sites, such as the Hollin Hill Landslide Observatory (HHLO) while having taken many years of research to establish proof-of-concept geophysics-supported slope-scale early warning (Fig. 3). This provides a blueprint for

establishing similar monitoring approaches more rapidly at future sites, streamlining the application of geophysics for LoLEWS (e.g., Holmes et al., 2020). Additionally, resource availability may limit the investment in to establishing all the aspects of geophysics for LoLEWS, in which case the HHLO case study provides a useful reference for which components

may be suited to other sites with specific and unique requirements. It is clear that, as observed in currently operating LoLEWS, an interdisciplinary approach is necessary in order to understand, integrate and exploit the many data streams feeding in to a

220 monitoring system. Furthermore, research in the field of geophysical monitoring can benefit from developments in other areas of study, e.g., machine learning, equipment manufacturing, signal processing and smart sensor networks.

## 4 Conclusions

Geophysical images, in particular 2D and 3D resistivity and seismic images, bring benefits to the establishment of LoLEWS, but are currently underutilised. Their ability to acquire high spatial resolution data and produce slope-scale models of the

225 subsurface, from reconnaissance studies to ground model development and the deployment of geophysical monitoring systems, makes them well suited to provide information for establishing and delivering LoLEWS. Resistivity monitoring is the most developed geophysical imaging method available for integrating in to LoLEWS. However, developments in DAS and nodal seismic arrays, which can provide hundreds to thousands of near-continuous seismic recording channels at comparable spatial resolutions to resistivity monitoring systems, will provide opportunities to acquire seismic velocity models at the same or

230 higher spatiotemporal resolutions as existing resistivity systems.

To translate geophysical measurements to slope-scale geotechnical or geomechanical models, and in turn use these in slope stability modelling activities, requires the use of petrophysical relationships. This is a key step in integrating geophysics in to all components of developing LoLEWS. Utilising data from slope-scale geophysical monitoring systems in this way can

introduce subsurface data to LoLEWS at a spatiotemporal resolution that would otherwise not be practicable to acquire by other intrusive or remote sensing methods. Whilst geophysical data eliminate some uncertainties, for example, by providing knowledge of the subsurface structure of a landslide at the slope-scale, other aspects of their use contain their own specific uncertainties (Fig. 1). The translation of geophysical models from their directly measured (geophysical) property to a linked geotechnical property provides three main benefits: i) the imaging of geotechnical properties and processes at a spatial

resolution that would be impracticable to replicate using individual sensors or remote sensing observations; ii) the provision of slope-scale, discretised inputs for physical- and process-based slope stability models; iii) greater understanding of data that have been translated from a technical geophysical measurement to a more universal engineering property which is more relevant to slope stability assessment (such as resistivity to moisture content or soil suction). The use of petrophysical relationships to translate geophysical models is particularly powerful when used as a monitoring tool. As laboratory

measurements are able to simulate a range of field conditions, and once petrophysical relationships are established, time-lapse field geophysical data can be rapidly translated to field-scale models of geotechnical and geomechanical properties. This unlocks possibilities for dynamic slope-scale modelling of stability using near-real-time field data at very high spatial resolutions.

## Code availability

No code was used in the production of this manuscript.

## Author contribution

The conceptualisation, formal analysis, visualisation and writing – original draft preparation were undertaken by J. S. Whiteley. All authors were responsible for writing – reviewing and editing. Supervision was provided by J. M. Kendall and J. E. Chambers.

## Competing interests

The authors have no competing interests to declare.

## Acknowledgements

This work was funded by a NERC GW4+ UK Doctoral Training Partnership Studentship (Grant NE/L002434/1), the BGS University Funding Initiative (S337), ACHILLES (UK Engineering and Physical Sciences Research Council (EPSRC) grant number EP/R034575/1) and is part of the 'Multi-observational data for linked modelling methodology development' activity of the NERC-funded Enhancing Resilience to Landslide Hazard project (NEE7680S). Jim Whiteley, Arnaud Watlet and Jonathan Chambers publish with the permission of the Executive Director, British Geological Survey (UKRI-NERC). All content generated as part of this work is copyright of British Geological Survey © UKRI 2020/ The University of Bristol 2020.

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

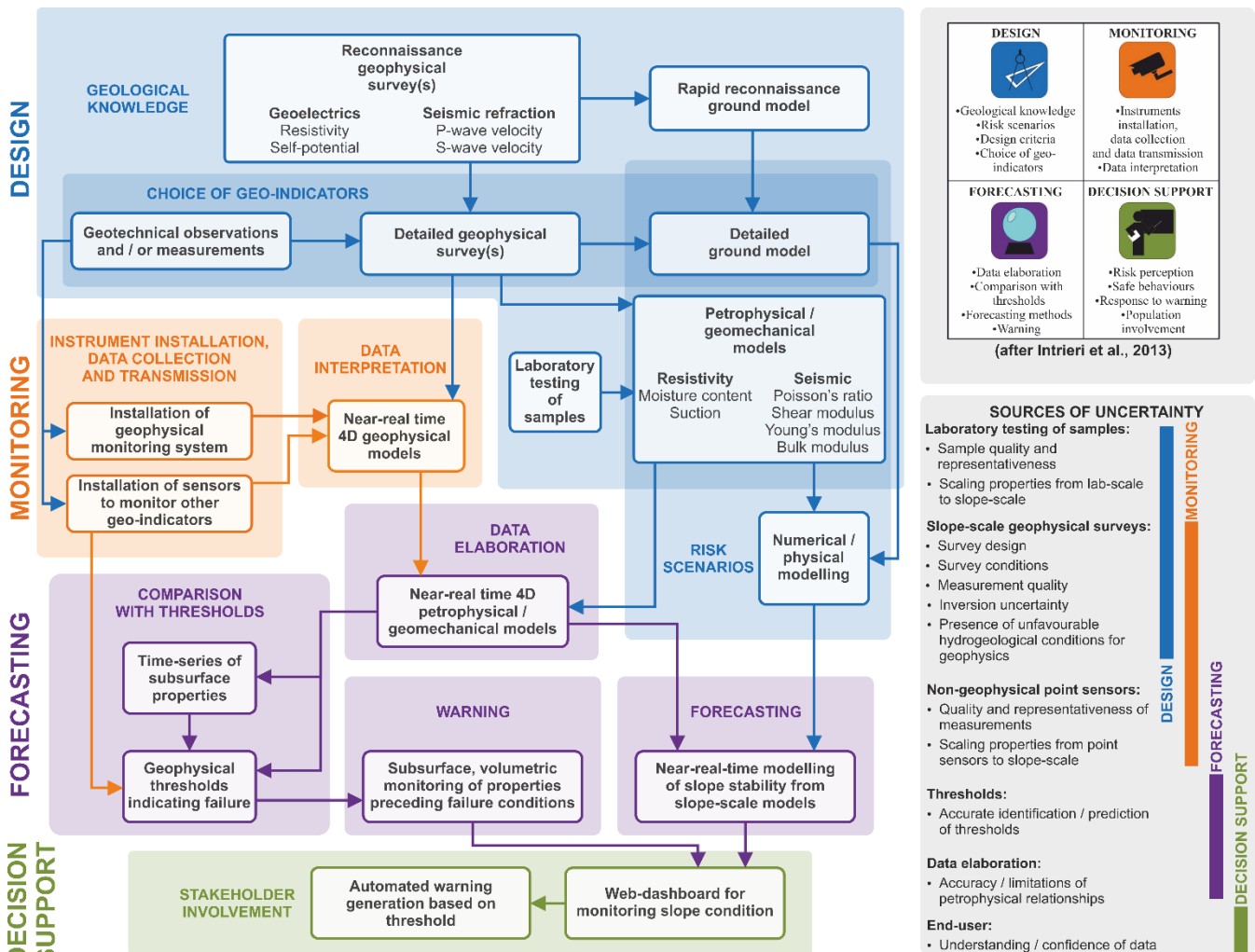

**Figure 1: A conceptual workflow illustrating the role of geophysics, supported by geotechnical observations and laboratory measurements, in establishing LoLEWS for landslides triggered by increases in ground moisture conditions. The uncertainties associated with different activities are also identified. Inset shows a modified version of the framework proposed by Intrieri et al. (2013), where we use the term 'decision support' instead of 'education', and 'stakeholder involvement' instead of 'population involvement'. 'Instrument installation', 'data collection' and 'data transmission' are considered as a single monitoring sub-component. The processing and interpretation (i.e., 'data interpretation') of geophysical data is considered a monitoring activity, and the integration and translation of geophysical data (i.e., 'data elaboration') is considered a forecasting activity.**

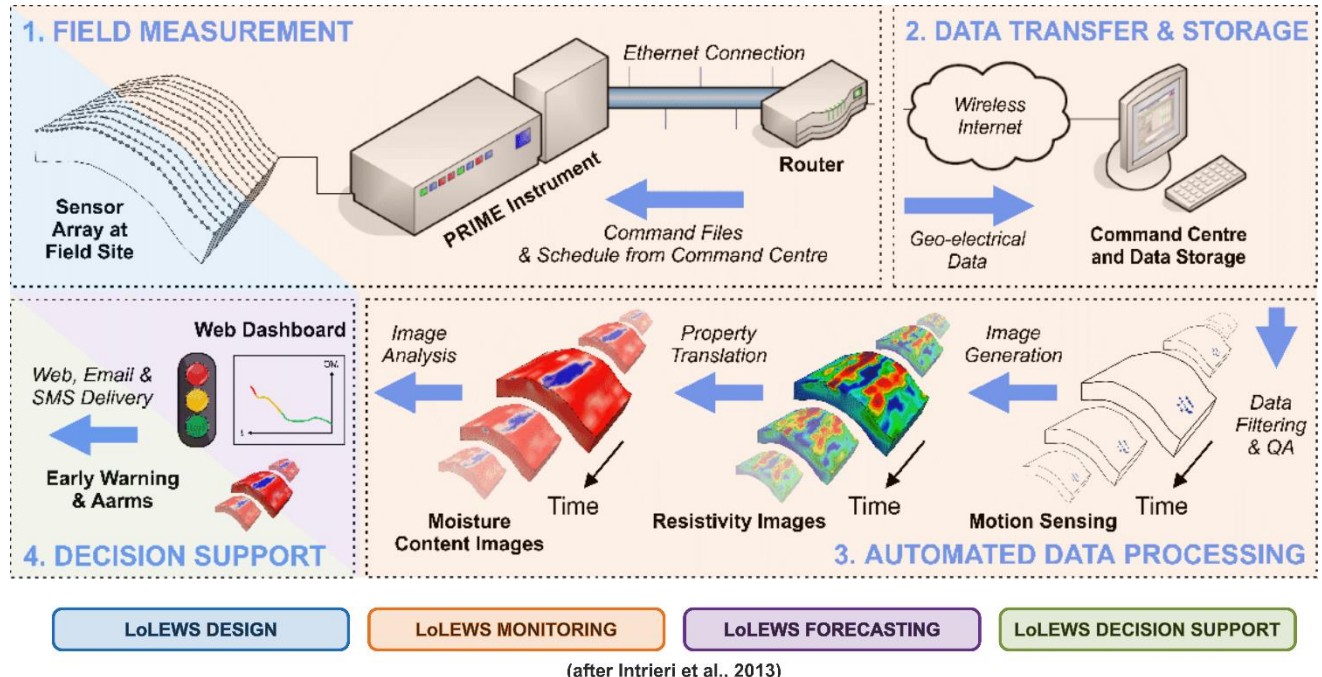

**Figure 2: The PRIME system workflow for handling resistivity monitoring data, demonstrating the acquisition, transmission, filtering, processing, translation and dissemination of geophysical models for slope failure early warning; the components of establishing LoLEWS are colour-coded to each stage (after Intrieri et al., 2013). Modified from Holmes et al. (2020).**

330

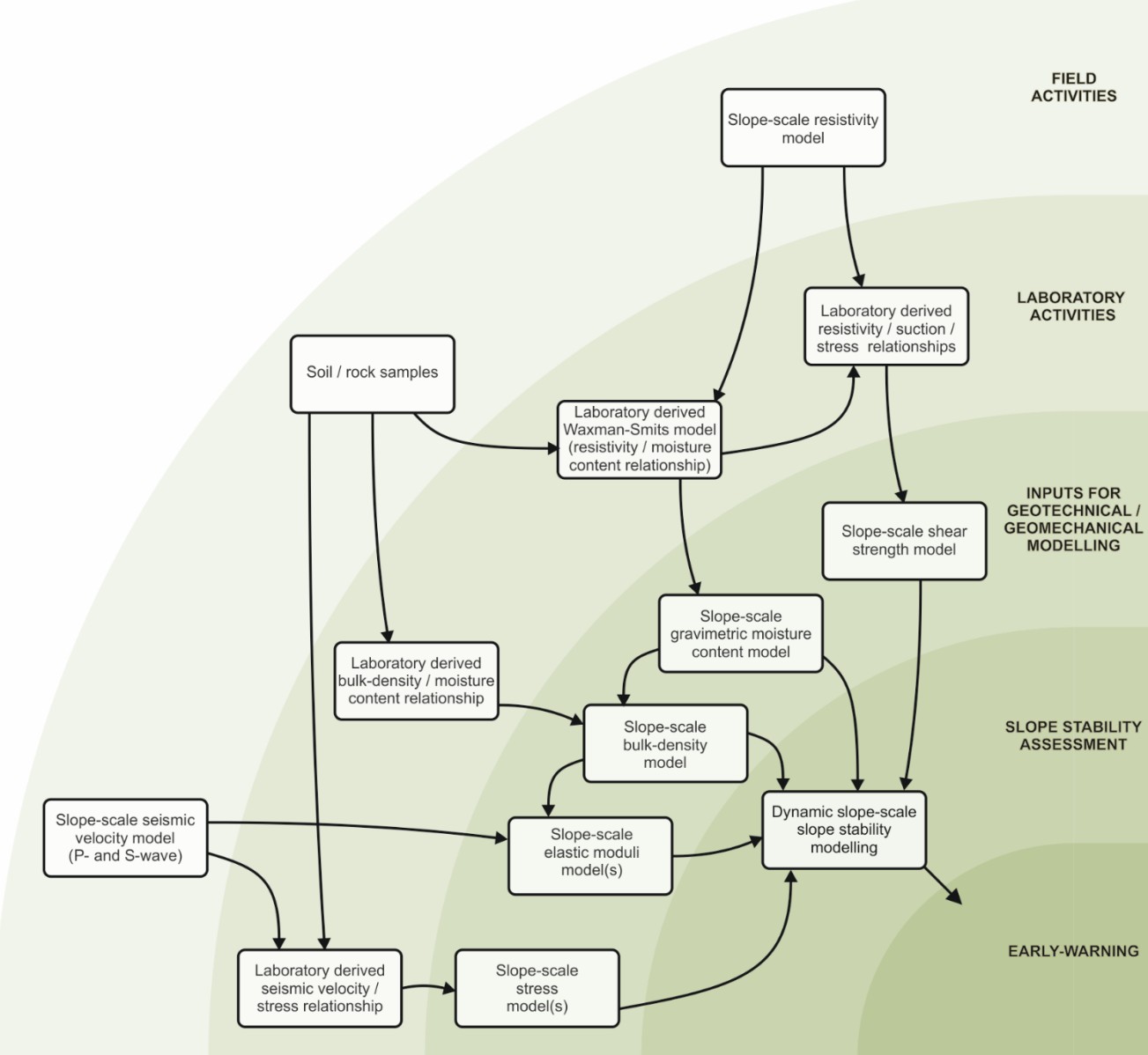

**Figure 3: A proposed roadmap from geophysical and geotechnical field measurements to landslide early-warning, by using laboratory measurements to transform geophysical data for slope-scale modelling, which has been developed for the Hollin Hill Landslide Observatory (HHLO) in the UK.**

335