# Peer review of "Brief communication: The role of geophysical imaging in local landslide early warning systems"

_Natural Hazards and Earth System Sciences, 2021_

## Author Response (AR1)

**Reviewer comments and author responses**

**Reviewer comment (RC1)**

The study provides a conceptual workflow for the integration of geophysical imaging in local landslide early warning systems (LoLEWS). Increased spatio-temporal resolution of geophysical data in combination with corresponding laboratory-based transformations can contribute to reducing uncertainties commonly associated with geological conditions on local scales and improve landslide forecasting. The authors are also encouraged to revise the paper and address the following comments:

RC1-1.   Please clarify if the conceptual framework for LoLEWS is limited to landslides triggered by changes in groundwater conditions.

RC1-2.   Quantitative analyses of slope failures and landslide forecasting with physical-based landslide prediction models often require knowledge on shear strength of soil. The relationship between geophysical data and shear strength properties of soil is not clear. Will the shear strenght properties be derived from classical tests (e.g., direct shear and triaxial tests) and enriched with geophysical data or directly from geophysical data? In Figure 3, the slope-scale shear streght model is derived from laboratory resistivity/suction/stress relationships and laboratory derived Waxman-Smith model. Please clarify if the laboratory tests also include shear strenght tests and if they are sufficient to describe shear strenght properties of soil.

RC1-3.   Although geophysical data can contribute to reducing uncertinties in in geological conditions, it is unlikely that uncertainties in geological properties will be completely eliminated. In addition to the uncertainties due to heterogeneous geological conditions, uncertainties will arise in, among others, the process of transforming geophysical data to geological parameters and landslide forecasting. The conceptual framework in Figure 1 and the roadmap in Figure 3 do not explicitly include steps or methodologies for dealing with uncertainties. Please clarify if a strategy for dealing with uncertainties is envisioned in the conceptual workflow.

RC1-4.   Is the technology necessary for collecting geophysical data suitable for being installed in very steep and remote areas, without access to power or internet and subjected to harsh weather conditions, which are typically encountered in deployments of landslide monitoring systems?

**Author response (AC1) to reviewer comment (RC1)**

Thank you for your comments (RC1) on our manuscript, which comprise a series of considered clarifications and additions which will be incorporated in to a revised version. In response to your comments:

35      AC1-1.      We will clarify that this framework applies primarily to landslides triggered by changes in groundwater conditions.

      AC1-2.      There are two approaches to estimating the shear-strength from geophysical data as outlined in Figure 3. As identified in the comment, one approach is to use ERT; in this approach, the ERT data are transformed to moisture

40      content using the (laboratory-derived) Waxman-Smits relationship, then, the moisture content is related to soil suction (Fredlund et al., 2011) before being transformed to shear strength (Vanapalli et al., 1996). A second approach is to use direct laboratory measurements of shear-strength and shear-wave velocity and derive a relationship between the two properties. This can be achieved using direct laboratory measurements of shear-strength and shear-wave velocity (e.g., using direct simple shear testing and bender element measurements) and/or field measurements (using shear-

45      vanes and field seismic measurements) (see Trafford and Long, 2020) or using a combination of database data and field measurements (see L'Heureux and Long, 2017). The difference between the means of estimating shear-strength (i.e., ERT data transforms and seismic laboratory and field relationships) and how they differ is not made clear in the manuscript, and further clarification can be added in revision.

50      AC1-3.      Our central message in this brief communication is that the addition of geophysical instrumentation in establishing and operating LoLEWS provides subsurface information at spatiotemporal scales that cannot be practically replicated using existing approaches. Geophysical approaches have the potential to provide spatially and temporally rich information in areas and/or volumes of the slope for which there would otherwise be no information at all. Hence, the addition of these geophysical data can help to reduce the overall uncertainties in quantifying

55      destabilising hydrogeological processes operating at the slope-scale. However, while we agree that the inclusion of geophysical data to LoLEWS will reduce, rather than eliminate, uncertainties surrounding geological properties, we also recognise that the use of geophysical and laboratory-based transforms will in turn introduce some new uncertainties to a LoLEWS system. We have not focused on uncertainties within this framework, as we aim to present a broad-scale route toward integration and inclusion of geophysical techniques in new and existing LoLEWS, and the

60      uncertainties surrounding these are highly site-specific, and would need to be understood in a local context. While this brief communication does not have the scope to discuss in detail the propagation of uncertainties (as this is still very much an open question in research), we recognise that acknowledging and understanding uncertainties is a crucial part in establishing LoLEWS. Therefore, we will update the conceptual workflow in Figure 1 to include sources of uncertainty that must be considered, and will include a brief subsection outlining these sources of

65      uncertainty.

      AC1-4.      Technological and hardware advances mean that geophysical systems are increasingly able to be installed in remote and difficult terrain (see Whiteley et al., 2019 and references therein). Examples cited in the manuscript and

elsewhere (the limit on the reference count for Brief Communications in NHESS preclude inclusion of many

70 examples) include deployments in "difficult" environments, for example: i) where mains power is not accessible, and local power has been generated by wind, solar or fuel cells (Uhlemann et al., 2017), ii) cellular networks have needed to be established in order to transmit data (Uhlemann et al., 2017), iii) equipment has had to be carried by hand over rough terrain rather than by vehicle access (Uhlemann et al., 2017), and iv) equipment has been subjected to harsh climatic conditions including annual freeze-thaw cycles in temperate environments (Holmes et al., 2020), monsoon

75 conditions (Watlet et al., 2019), permafrost conditions (Uhlemann et al., 2021) and arctic studies (Cimpoiasu et al., 2021). Generally, the major limitations on installation of these systems are related to access, rather than shortcomings in the equipment. Another reviewer (RC2) has raised a similar point regarding cost and robustness, and we will include text to emphasise the points raised above, and in response to their comment.

80 **Reviewer comment (RC2)**

The submission is a brief conceptual paper that can be seen as a "case" for the use of time-lapse geophysical surveys in local early warning systems (LoLEWS) for weather-induced landslides. The arguments, clearly expressed and well framed in the context of reference literature, are centered around three well-designed and self-explanatory figures, highlighting the role that geophysical field measurements have in the workflow of activities needed for landslide early warning. The paper is almost

85 ready for being published as is, yet the following minor revisions are suggested.

RC2-1.    Some issues that are worth being considered in this discussion are the costs and robustness of the geophysical deployments, and the effectiveness of geophysical surveys in relation to different types of soils and landslides.

90 RC2-2.    Always introduce the meaning of acronyms when they are first used (HHLO, DAS).

RC2-3.    The first paragraph of conclusions does not derive from previous comments in the article. Thus, it is more appropriate to move it at the end of the conclusions.

95 **Author response (AC2) to reviewer comment (RC2)**

Thank you for your comments (RC2) on our manuscript, which will be incorporated in to a revised version. In response to your comments:

AC2-1.    Another reviewer (RC1) has raised a similar point regarding the ability of equipment to be deployed in
100    remote and difficult to access areas, and we will include further information on these aspects to emphasise the cost-benefit and increasing robustness of these systems. Please see reply to comment (AC1) for details.

AC2-2.    We will properly introduce the acronyms in the manuscript.

105    AC2-3.    The conclusion will be reorganised into a more logical format.

**References for author comments**

Cimpoiasu, M.O., Meldrum, P., Harrison, H., Chambers, J., & Kuras, O. (2021). "Year-round 4D electrical resistivity imaging to monitor the hydrodynamics of deglaciated Arctic soils", SEG: Application of Proximal and Remote Sensing Technologies for Soil Investigations Symposium, 16-19 August 2021.

Fredlund, D. G. F. G., Sheng, D. & Zhao, J. 2011. Estimation of soil suction from the soil-water characteristic curve. Canadian Geotechnical Journal, 48, 186-198.

Holmes, J., Chambers, J., Meldrum, P., Wilkinson, P., Boyd, J., Williamson, P., Huntley, D., Sattler, K., Elwood, D., Sivakumar, V., Reeves, H. & Donohue, S. 2020. Four-dimensional electrical resistivity tomography for continuous, near-real-time monitoring of a landslide affecting transport infrastructure in British Columbia, Canada. Near Surface Geophysics, 18, 337-351.

L'Heureux, J.-S. & Long, M. 2017. Relationship between Shear-Wave Velocity and Geotechnical Parameters for Norwegian Clays. Journal of Geotechnical and Geoenvironmental Engineering, 143, 04017013.

Trafford, A. & Long, M. 2020. Relationship between Shear-Wave Velocity and Undrained Shear Strength of Peat. Journal of Geotechnical and Geoenvironmental Engineering, 146, 04020057.

Uhlemann, S., Chambers, J., Wilkinson, P., Maurer, H., Merritt, A., Meldrum, P., Kuras, O., Gunn, D., Smith, A. & Dijkstra, T. 2017. Four-dimensional imaging of moisture dynamics during landslide reactivation. Journal of Geophysical Research: Earth Surface, 122, 398-418.

Uhlemann, S., Dafflon, B., Peterson, J., Ulrich, C., Shirley, I., Michail, S. & Hubbard, S. S. 2021. Geophysical Monitoring Shows that Spatial Heterogeneity in Thermohydrological Dynamics Reshapes a Transitional Permafrost System. Geophysical Research Letters, 48, e2020GL091149.

Vanapalli, S., Fredlund, D., Pufahl, D. & Clifton, A. 1996. Model for the prediction of shear strength with respect to soil suction. Canadian geotechnical journal, 33, 379-392.

Watlet, A., Thirugnanam, H., Singh, B., Kumar, N. M., Brahmanandan, D., Swift, R. T., Inauen, C., Meldrum, P., Uhlemann, S. & Wilkinson, P. B. Deployment of an electrical resistivity monitoring system to monitor a rainfall-induced landslide (Munnar, India). AGU Fall Meeting Abstracts, 2019. H14A-03.

Whiteley, J. S., Chambers, J. E., Uhlemann, S., Wilkinson, P. B. & Kendall, J. M. 2019. Geophysical Monitoring of Moisture-Induced Landslides: A Review. Reviews of Geophysics, 57, 106-145.

---

## Author Response (AR3)

**Reviewer comments and author responses**

**Reviewer comment (RC1)**

The study provides a conceptual workflow for the integration of geophysical imaging in local landslide early warning systems (LoLEWS). Increased spatio-temporal resolution of geophysical data in combination with corresponding laboratory-based transformations can contribute to reducing uncertainties commonly associated with geological conditions on local scales and improve landslide forecasting. The authors are also encouraged to revise the paper and address the following comments:

RC1-1    Please clarify if the conceptual framework for LoLEWS is limited to landslides triggered by changes in groundwater conditions.

AC1-1    We have clarified this point in the introduction when introducing Fig. 1 (l.19 and l. 41) and also in the figure caption for Fig. 1

RC1-2    Quantitative analyses of slope failures and landslide forecasting with physical-based landslide prediction models often require knowledge on shear strength of soil. The relationship between geophysical data and shear strength properties of soil is not clear. Will the shear strenght properties be derived from classical tests (e.g., direct shear and triaxial tests) and enriched with geophysical data or directly from geophysical data? In Figure 3, the slope-scale shear streght model is derived from laboratory resistivity/suction/stress relationships and laboratory derived Waxman-Smith model. Please clarify if the laboratory tests also include shear strenght tests and if they are sufficient to describe shear strenght properties of soil.

AC1-2    We have clarified the different means of estimating shear-strength (between l.169 and l.175) specifically with the re-wording and expansion of the following sentence: "*Alternatively, shear-strength and shear-wave velocity can be measured in the laboratory (e.g., using direct simple shear testing and bender element measurements) or in the field (e.g., using shear vanes and field seismic measurements) to derive a relationship between the two properties (e.g., Trafford and Long, 2020).*"

RC1-3    Although geophysical data can contribute to reducing uncertinties in in geological conditions, it is unlikely that uncertainties in geological properties will be completely eliminated. In addition to the uncertainties due to heterogeneous geological conditions, uncertainties will arise in, among others, the process of transforming geophysical data to geological parameters and landslide forecasting. The conceptual framework in Figure 1 and the roadmap in Figure 3 do not explicitly include steps or methodologies for dealing with uncertainties. Please clarify if a strategy for dealing with uncertainties is envisioned in the conceptual workflow.

35    AC1-4   We have updated Fig.1 to reflect the major sources of uncertainty when considering geophysical imaging for LoLEWS, and have coded these uncertainties against their activity in the Fig.1 framework fro clarity.

We have also acknowledged the role of uncertainty when introducing Fig.1 in the text (l.47 – l.49), and have discussed this in the conclusions (l.235 – 237). We also include a summary of the new information in Fig.1 at the end of Section 2 (l.79 – l.90):

40    *"The framework also highlights the major sources of potential uncertainty that require consideration, including: i) laboratory testing of samples (e.g., quality of samples and how well samples reflect the heterogeneity of a geological formation) and issues scaling laboratory measurements to the slope-scale, ii) geophysical surveys, including the design of the survey (e.g., equipment used, data coverage and data resolution), the survey conditions (e.g., signal-to-noise ratio impacting measurement quality), uncertainties surrounding the inversion process (e.g., model sensitivity, fitting of geophysical models to measured*

45    *data, or the sensitivity of the inversion to regularisation constraints), and the presence of unfavourable conditions for geophysical surveying (e.g., buried low-velocity or high conductivity zones inhibiting accurate seismic and geoelectrical measurements respectively), iii) the accuracy, precision and representativeness of non-geophysical point sensors used to supplement geophysical monitoring systems, and issues surrounding scaling localised point-sensor measurements to the slope-scale, iv) the identification of geophysical thresholds at which critical slope conditions are reached, v) the limitations,*

50    *particularly in the ranges of co-sensitivity, of petrophysical relationships, and vi) the confidence of end-users to incorporate sources of geophysical data in to decision making processes surrounding slope-scale early warning."*

RC1-4   Is the technology necessary for collecting geophysical data suitable for being installed in very steep and remote areas,
55    without access to power or internet and subjected to harsh weather conditions, which are typically encountered in deployments of landslide monitoring systems?

AC1-4   It is not possible within the scope of the Brief Communication to list all of the examples given in the public discussion stage of the review of this manuscript, however, we have summarised those points in the following sentence (l.144 – l.149):
60    *"The low-cost modular and robust construction of monitoring systems such as PRIME, combined with their low-power consumption and optimised acquisition and data telemetry schedules make them adaptable for installing in harsh environments and difficult terrain, for example, where access may only be possible on foot (see Holmes et al., 2020; Whiteley et al., 2019 and references therein)."*

65

**Reviewer comment (RC2)**

The submission is a brief conceptual paper that can be seen as a "case" for the use of time-lapse geophysical surveys in local early warning systems (LoLEWS) for weather-induced landslides. The arguments, clearly expressed and well framed in the

context of reference literature, are centered around three well-designed and self-explanatory figures, highlighting the role that geophysical field measurements have in the workflow of activities needed for landslide early warning. The paper is almost ready for being published as is, yet the following minor revisions are suggested.

RC2-1   Some issues that are worth being considered in this discussion are the costs and robustness of the geophysical deployments, and the effectiveness of geophysical surveys in relation to different types of soils and landslides.

AC2-1   We have added a sentence summarising the points raised in the public discussion stage of the review of this manuscript, which addresses these points (l.146 – l.149): "*The low-cost modular and robust construction of monitoring systems such as PRIME, combined with their low-power consumption and optimised acquisition and data telemetry schedules make them adaptable for installing in harsh environments and difficult terrain, for example, where access may only be possible on foot (see Holmes et al., 2020; Whiteley et al., 2019 and references therein).*"

RC2-2   Always introduce the meaning of acronyms when they are first used (HHLO, DAS).

AC2-2   The first instances of these acronyms in the manuscript have now been correctly identified and expanded (HHLO, l.213; DAS, l.140)

RC2-3   The first paragraph of conclusions does not derive from previous comments in the article. Thus, it is more appropriate to move it at the end of the conclusions.

AC2-3   Some of the information related to identifying collaborative partners, maintaining observatories for research and maintaining an inter-disciplinary approach have been moved to the preceding section 3 (The future of geophysics for LoLEWS). We have then re-organised the remaining text in to a more logical order that better represents the flow of the manuscript.

---

## Author Response (AR4)

**Editor comments and author responses**

Thank you for your comments on our manuscript. Please find comments below.

5    On a final read through, we have made some very minor adjustments to the text to improve the flow and clarity. We hope these very minor changes are acceptable. These are (changes in italics):

- amended one line of the second paragraph of the introduction (line 31) from "by characterising the subsurface" to *"(i.e., characterisation of the subsurface)"*.

- changed "conceptual model creation" to *"conceptual site model"* in the first line (line 108) of Section 2.1.1 (Geological knowledge).

10

- added an instance of a reference (Whiteley et al., 2021) which was already in the reference list, to paragraph 1 of Section 2.2.2 (Data interpretation) (line 154).

- Split the long sentence at line 212 which read "Additionally, the work undertaken at natural observatory sites, such as the Hollin Hill Landslide Observatory (HHLO) while having taken many years of research to establish proof-of-

15    concept geophysics-supported slope-scale early warning (Fig. 3), provide a blueprint for similar developments more rapidly at future sites, streamlining the application of geophysics for LoLEWS (e.g., Holmes et al., 2020)."  This now reads "*Additionally, the work undertaken at natural observatory sites, such as the Hollin Hill Landslide Observatory (HHLO) while having taken many years of research to establish proof-of-concept geophysics-supported slope-scale early warning (Fig. 3). This provides a blueprint for establishing similar monitoring approaches more rapidly at*

20    *future sites, streamlining the application of geophysics for LoLEWS (e.g., Holmes et al., 2020).*"

**Editor comment**

EC3-1    is the term "ground moisture" an appropriate term (i.e. does not get confused with soil moisture)? Perhaps you may use groundwater, but I am not sure that this is you intention. I am ok if you leave "ground moisture", but please make a check

25

AC3-1    We prefer the term "ground moisture", as it doesn't distinguish between water held in soil or rock environments, although we acknowledge that "soil moisture" is a more common term. We have added one short sentence in the introduction (line 20) to clarify this to the reader, and amended two instance of "soil moisture" later in the manuscript to "ground moisture" to be consistent (lines 182 and 183) . We avoid the term "groundwater" as there is a strong association between this term and

30    water beneath the water table, which is not the only source of water we aim to monitor.

EC3-2    in figure 1 add space between 'with' and 'thresholds' (FORECASTING)

AC3-2    Amended.

i